# Relationship between Built-Up Environment, Air Pollution, Activity Frequency and Prevalence of Hypertension—An Empirical Analysis from the Main City of Lanzhou

**DOI:** 10.3390/ijerph20010743

**Published:** 2022-12-31

**Authors:** Haili Zhao, Minghui Wu, Yuhan Du, Fang Zhang, Jialiang Li

**Affiliations:** 1College of Geography and Environmental Science, Northwest Normal University, Lanzhou 730070, China; 2Key Laboratory of Resource Environment and Sustainable Development of Oasis, Lanzhou 730070, China

**Keywords:** built environment, hypertension, pathways of action, multilevel generalized structural equation model, healthy public policy

## Abstract

In the process of promoting the strategy of a healthy China, the built environment, as a carrier of human activities, can effectively influence the health level of residents in the light of its functional types. Based on the POI data of four main urban areas in Lanzhou, this paper classifies the built environment in terms of function into four types. The association between different types of built environments and the prevalence of hypertension was investigated by using the community as the study scale, and activity frequency, air pollution and green space were used as mediating variables to investigate whether they could mediate the relationship between built environments and hypertension. The results indicate that communities with a high concentration of commercial service facilities, road and traffic facilities and industrial facilities have a relatively high prevalence of hypertension. By determining the direct, indirect and overall effects of different functional types of built environment on the prevalence of hypertension, it was learned that the construction of public management and service facilities can effectively mitigate the negative effects of hypertension in the surrounding residents. The results of the study contribute to the rational planning of the structure of the built environment, which is beneficial for optimizing the urban structure and preventing and controlling chronic diseases such as hypertension.

## 1. Introduction

With the rapid development of urbanization in China, people’s living standards are constantly improving, living environment is constantly changing and the prevalence of chronic noncommunicable diseases is increasing year by year, which has become the main factor threatening health. According to the statistics of the World Health Organization in 2021, the mortality rate of chronic noncommunicable diseases caused by cardiovascular diseases was the highest, and the death toll reached 17.9 million in 2019, accounting for 32% of the total deaths in the world. Among them, hypertension was considered as one of the most dangerous factors leading to cardiovascular diseases [1]. However, overreliance on transportation, lack of physical activity, high stress in life and work, air pollution and ecosystem imbalance will all cause the increase of its prevalence [2,3,4,5]. According to the strategic plan of the Fifth Plenary Session of the 18th CPC Central Committee, the State Council promulgated the Outline of “Healthy China 2030”, aiming at promoting the process of building a healthy China and improving people’s health. Therefore, it is very important for public health to study the influencing factors of hypertension and their internal relations.

As the spatial background of urban residents’ activities, the built-up environment can have a significant impact on residents’ health, and it has the advantages of long-term effect, extensive scope of action and high feasibility of optimization and improvement, which leads to more and more studies focusing on its impact on residents’ health from the built-up environment [6,7]. Binay and others started from two aspects: walking ability and park availability and concluded that the built environment had a positive effect on human health by improving residents’ awareness and frequency of activities 4. A study from Suzhou, China proved that the functional density, mixing degree, road network and traffic station density and greening rate of built environment exerted positive or negative impacts on residents’ health [8]. Among the identified pathways of a series of factors affecting the prevalence of hypertension, the built environment is first regarded as the carrier of human activities, and its facility density, function type, road density, integration degree and other indicators have become the standards for evaluation whether the built environment can enhance the intensity of residents’ activities. By accumulating social capital, reducing crime rate and improving residents’ happiness, residents’ travel frequency is increased, with various travel modes and leisure activities, and thus the prevalence of chronic noncommunicable diseases such as hypertension, diabetes, respiratory diseases and obesity is reduced [9,10,11,12]. For example, Cao Xinyu and others believed that the high-density built environment greatly reduced the time and space distance between destinations and creates green travel conditions for residents to improve physical activity and reduce health risks [13]; On the contrary, the expansion of urban built-up areas led to an increase in commuting time, which made residents prefer non-green modes of travel such as private cars, resulting in a decrease in moderate and high-intensity physical activity, which, in turn, increased BMI, obesity rate and the prevalence of chronic noncommunicable diseases [14,15,16]. Secondly, air pollution is an important factor endangering human health, and the built environment does have a certain impact on air quality. For example, air pollution was more serious in places where a large amount of domestic and industrial waste gas in dense roads and factories was emitted than other places [17,18]. The current research combines air pollution exposure with health and analyzes the influence of air pollution concentration and exposure duration on the prevalence and mortality of chronic noncommunicable diseases. Some studies also deeply discuss the differences of different environmental exposures on different individual exposures, such as age differences, regional differences and individual prevalence [17,19] Xu Yanting and others believed that physical activity was not the leading factor affecting health, and the health risks caused by air pollution exposure should be considered at the same time [18]; Li Zhiguang used the fixed effect model to verify that air pollution did have a negative impact on health, and there were differences in the degree of health inhibition among different groups [20]; A medical study started with chronic noncommunicable diseases such as myocardial infarction, cerebrovascular diseases, hypertension and diabetes and confirmed that air pollution did increase the risk of diseases [21]. Finally, green space can alleviate diseases. With a higher overall green degree and human health in the built environment, the incidence of chronic noncommunicable diseases [22,23,24,25,26] became lower. Green facilities such as parks and scenic spots could reduce the incidence of chronic diseases such as hypertension and diabetes by reducing the concentration of air pollutants and promoting residents’ physical exercise and play a positive role in promoting health [27]. Aerts et al. found that there was a negative correlation between green space and the probability of suffering from chronic noncommunicable diseases by judging human health from drug sales [28]. However, a study from Madrid took different buffer zones as the evaluation scope and explored that the green index reduced the risk of cardiovascular disease [29]. Hypertension is an important risk factor affecting residents’ health, so it is of great practical significance to explore its influencing factors. At present, on studying the influence of environmental factors on hypertension, although it has been confirmed that the built environment does affect residents’ health by influencing the surrounding greening degree, air pollution concentration or residents’ activity intensity, researchers often only pay attention to the effect of a single intermediary factor or a single path on hypertension and rarely delve into the impact of the combination of multiple influencing factors on hypertension [29,30,31]. Moreover, the current research is mostly started on a large scale [15] and less studied on the impact of built-up environment on hypertension on a small scale. In addition, the existing studies pay little attention to the influence of built-up environmental function types on the prevalence of hypertension. Therefore, this study subdivides the functional types of built-up environment, taking the community as the basic research scale, and analyzes how the interwoven paths mediate the relationship between the built-up environment with different functional types and the prevalence of hypertension.

In Lanzhou, as the capital of Gansu Province and the gathering place of heavy industry, the functional types of built environment are complex, with poor air quality. According to previous research results, as a place for daily activities, the impact of its environmental conditions on residents’ health is second only to the impact of individual characteristics on health, and there is an interactive relationship between residents’ travel activities and their surrounding built-up environment and air quality. The difference of built-up environment around the community will affect the environmental conditions and residents’ living conditions from many aspects, thus affecting chronic diseases such as hypertension. Therefore, combined with the existing achievements, taking the main urban areas of Lanzhou as the research areas, this paper divides the built-up environment according to the land use type and explores the relationship between the built-up environment of different functional types and the prevalence of hypertension based on the community scale, aiming at providing policy suggestions for the rational planning of urban facility types and control and prevention of hypertension.

## 2. Study Area, Data and Methods

### 2.1. General Situation of Study Area

Lanzhou City (36°03′ N, 103°40′ E), the capital of Gansu Province, is the political, economic and cultural center of the whole province. It is located in the geometric center in China’s land map, with mountain ranges along the north and south and the Yellow River running through the urban area from the east to the west. It is a transitional area from the Qinghai–Tibet Plateau to the Loess Plateau. Most areas in the territory are basins and hills covered by loess, with an average altitude of over 1500 m. The whole city now has jurisdiction over Chengguan, Qilihe, Xigu, Anning and Honggu Districts and Yongdeng, Yuzhong and Gaolan Counties. Up to 2020, the total land area of the city was 13,085.6 square kilometers, and the total registered population was 3,222,800, including 2.1 million urban population. The study area of this paper includes 50 streets in four main urban areas of Lanzhou (Chengguan District, Anning District, Qilihe District and Xigu District), which are divided into 360 communities according to the national statistical zoning code and urban–rural zoning code in 2021 (Figure 1). Lanzhou is a gathering place of heavy industries in the western region. Limited by river valley topography, the air circulation is not good enough. In addition, the local climate is special, and the inversion layer is thick. Therefore, it is not easy to let the traffic, industrial and domestic waste gas diffuse and let fresh air in, which has become the main reason for the poor air quality in Lanzhou.

### 2.2. Participants’ Caracteristics

The data of hypertension in Lanzhou in 2020 came from four main urban areas (Chengguan District, Anning District, Qilihe District and Xigu District), According to the data of blood pressure measurements collected from streets and community health service stations, 103,877 hypertensive patients (including systolic blood pressure ≥ 140 mmHg or diastolic blood pressure ≥ 90 mmHg) were screened from all cases of blood pressure measurements, including age, gender, height and weight, community address, systolic blood pressure, diastolic blood pressure, BMI and other information. The average population density of 360 communities was estimated according to the 1 km grid data of spatial distribution of the Chinese population released by Resources and Environmental Science and Data Center of the Chinese Academy of Sciences in 2020.

### 2.3. Built Environment Assessment

Compared with traditional land data, which is hard to obtain and update in time, POI data can more accurately measure the functional types of built environments [32,33]. Therefore, the built-up environmental data of the main Lanzhou City in 2020 required by this study were obtained by python crawler technology. Through preprocessing such as cleaning, comparison, spatial correction and eliminating outliers, POI that could not be obtained and incomplete information were deleted, and a total of 10,815 POI data were retained to represent the built-up environmental function types.

In previous studies on the prevalence of hypertension, it was found that different functional built-up environments had different effects on residents’ health [34]. Compared with other functional types of facilities, the built environment with public service, transportation, commercial and industrial functions was more likely to affect human health by changing residents’ living patterns and surrounding environmental conditions. Therefore, these facilities accounted for a heavier proportion of the impact of the built environment on health, and their research theories were also richer [4,5,18,35]. The purpose of this paper is to discuss the relationship between the built environment of different functional types and the prevalence of hypertension. Taking the Standards for Urban Land Classification and Planning and Construction Land implemented in 2012 for a reference, combined with the characteristics of heavy industries in Lanzhou, the collected POI data are classified into four categories: public management and service facilities, commercial service facilities, road and transportation facilities and industrial facilities. Among them, there are 10,592 public management and service facilities (3832 medical and health service facilities; 274 sports service facilities (gymnasiums, swimming pools and various ball venues, etc.); 5216 science, education and cultural service facilities; 882 public facilities and 388 scenic spots); 93,485 commercial service facilities (15,285 catering service facilities, 10,520 corporate enterprises, 37,096 shopping service facilities, 2769 financial and insurance service institutions, 5020 automobile maintenance and sales service facilities, 16,972 living service facilities, 2506 leisure service facilities, 307 sports service facilities (racetrack, golf, etc.) and accommodation service facilities); 3843 roads and transportation facilities (parking lots, bus stops, etc.) and 238 industrial facilities in total. Linked with 360 communities, the distribution density of four built-up environment types in each community is calculated.

### 2.4. Mediating Variable

Existing studies have identified the effects of air quality, physical activity and greening on blood pressure. Therefore, this paper considers three intermediaries at the regional level: air pollution, activity frequency and green space. Firstly, the average concentration of PM_2.5_ in each community in 2020 was calculated to assess the air pollution level. PM_2.5_ concentration data came from the Lanzhou Ecological Environment Bureau, which was obtained by downloading the daily updated environmental quality release information, summarized and sorted out after abnormal values or blank values were eliminated. Secondly, based on the comprehensive consideration of residents’ actual activities and the difficulty of data acquisition, Baidu thermal data at 6:00, 10:00, 14:00, 18:00 and 22:00 in a day were integrated, and working days and rest days in a week were selected as the average weekly activity frequency, and the activity frequency of residents in the community was evaluated by this data. Due to the lack of obtained data and other reasons, the monthly average thermodynamic value was calculated by interpolation of the data. Finally, the green space data was obtained by collecting Google Earth remote sensing images with a resolution of 30 m in August 2020 (the green space grows best), and the images were preprocessed by remote sensing interpretation, radiometric calibration and geometric correction, and the Normalized Differential Vegetation Index (NDVI) was calculated by ENVI5.3. These three mediators are set as continuous variables [36].

### 2.5. Research Method

#### 2.5.1. Spatial Autocorrelation Analysis

The spatial autocorrelation analysis can be divided into a global spatial autocorrelation analysis and local spatial autocorrelation analysis [37] according to the research objects. In this paper, the Moran index is used to evaluate whether the prevalence rate of hypertension has spatial aggregation [38]. Its basic principles are as follows:(1)$$I=\frac{n\sum_{i=1}^{n}\sum_{j=1}^{n}w_{i,j}z_{i}z_{j}}{s_{0}\sum_{i=1}^{n}z_{i}^{2}}$$
where *I* is Moran’s *I* index, and the value of *I* is between −1 and 1. A positive value is a positive correlation, while a negative value is a negative correlation. The greater the correlation is, the stronger the correlation is. $Z_{i}$ is the deviation between the attributes of elements i and the average value $\left(X_{i}-\bar{X}\right)$; $W_{i,j}$ is the spatial weight between elements i and j; n refers to the total number of elements and $S_{0}$ is the aggregation of all spatial weights. 

Local Getis-OrdGi* hot spot detection is used to judge the spatial distribution of cold hot spots in hypertension prevalence [39,40]. The calculation formula is as follows [41]:(2)$$\begin{array}{c} G_{i}^{*}=\frac{\sum_{j=1}^{n}W_{ij}x_{j}-W_{j}^{*}\dot{x}}{s\sqrt{\frac{\left[nS_{1i}-{\left(W_{i}^{*}\right)}^{2}\right]}{n-1}}},\;W_{i}^{*}=\sum_{j=1}^{n}W_{ij},\;S_{1i}=\sum_{j=1}^{n}W_{ij}^{2} \end{array}$$
where *S* is the standard deviation of the incidence of hypertension in the four main urban areas of Lanzhou, and the values of $G_{i}^{*}$ reflect the high and low values of cold hot spots. The greater the absolute value of $G_{i}^{*}$ is, the higher the aggregation degree of the observed values in this area, viewed as hot spots. That is, positive $G_{i}^{*}$ values are denoted as high-value aggregations, while negative $G_{i}^{*}$ values as low-value aggregations [42].

#### 2.5.2. Multilevel Generalized Structural Equation Model

A multilevel generalized structural equation model is suitable for constructing complex impact paths and analyzing the magnitude and direction of the impact of each path more accurately. This method can be applied when there are multiple measured variables in the study and the relationship between the variables needs to be analyzed. In view of the fact that the data has obvious nested structure (that is, the sample individuals are nested in the community), therefore this paper adopts hierarchical and multilevel generalized structural equation model (MGSEM) of four built environment types: public management and service facilities, commercial service facilities, road and transportation facilities and industrial facilities. To study the relationship between built-up environment and hypertension prevalence, three potential mediators (air pollution, activity frequency and NDVI) are adjusted by controlling covariates at sample individual and regional levels. Structural equation modeling can explore the relationship between built-up environment and hypertension by analyzing model paths and effect values. Various effects between independent variables and dependent variables can be represented by Direct Effect, Indirect Effect and Total Effect [43]. By testing the Red Pool Information Criterion (AIC) and Bayesian Information Criterion (BIC) to evaluate the fitting degree of the model, the sequence intermediary model is selected to study the influence effect of each path. Set the output attribute in Amos to calculate the mediation effect of the sequence mediation model. In this model (Figure 2), the built environment can affect hypertension through six paths: Path 1: built environment → PM_2.5_ concentration → hypertension; Path 2: built environment → PM_2.5_ concentration → activity frequency → hypertension; Path 3: built environment → NDVI → PM_2.5_ concentration → hypertension; Path 4: built environment → NDVI → activity frequency → hypertension; Path 5: built environment → NDVI → PM_2.5_ concentration → activity frequency → hypertension and Path 6: Built environment → activity frequency → hypertension. The three mediators interact with each other and have a common effect on hypertension [44].

Data were analyzed through three steps: (1) Modeling: MGSEMs of six built environments were fitted to explore the relationship between hypertension, built environment and three mediators (PM_2.5_ concentration, activity frequency and NDVI). (2) Judge that fitting degree of the model. (3) The direct and indirect effects of each pathway were calculated separately, and the relative contribution of each pathway to the overall effect of built environment on hypertension was explored. Amos 23 was used for statistical analysis to judge whether the pathway was significant according to *p* < 0.05.

## 3. Results and Analysis

### 3.1. Participants’ Characteristics

Table 1 shows the descriptive characteristics of hypertensive patients in four main urban areas and the distribution density of different built-up environment types. The density of the four built-up environment types is the highest in Chengguan District, which obviously exceeds the distribution density in other three districts. There are also significant differences in the number, sex, age, BMI, population density, PM_2.5_ concentration and thermodynamic value of hypertension patients among the four urban areas. Chengguan District has the largest number of patients, accounting for 54.11% of the total number of patients, Qilihe District with the highest population density accounts for 20.24%, and Anning District and Xigu District have fewer patients. Among all patients with hypertension, male samples account for 41.20%, and female samples account for 58.80%, slightly more than male samples. According to the age definition of the elderly population in the United Nations World Assembly on Ageing in 1982 [45], 65.84% of the patients are ≥60 years old, and the whole sample is aging.

### 3.2. Evaluation of Overall Matching Degree of Equation Model

The initial model is established by Amos23. In order to ensure the expected effect of the structural equation model, chi-square value, chi-square degree of freedom ratio (X^2^/df), mean square and square root of asymptotic residual (RMSEA) are calculated to verify the fitting degree between the data and the model. Table 2 lists the indicators for evaluating the adaptation degree of the overall model. After comparing with the adaptation standard, it is found that the other data fall within the recommended value range except that the chi-square degree of freedom ratio of public management and service facilities and commercial service facilities is slightly higher than the standard value. Therefore, it can be judged that the theoretical model established in this study is valid [46].

### 3.3. Spatial Distribution Pattern of Hypertension in the Main Urban Area of Lanzhou City

Moran’s I value of hypertension incidence in 2020 passed the significance test and Z value was greater than 2.58, which indicated that the incidence of hypertension showed spatial agglomeration characteristics.

By using local Getis-OrdGi* hot spot detection to judge the local spatial agglomeration of hypertension prevalence, according to the research results, it is found that both hot spots and cold spots exist in the region, with hot spots significantly concentrated in the west of Chengguan District, the east of Anning District and the east of Qilihe District, while cold spots significantly concentrated in the west of Xigu District and the northwest of Anning District, and the prevalence of hypertension in other areas shows random distribution characteristics. Hypertensive patients are mostly concentrated in urban areas, and there are relatively few hypertensive patients in the periphery of urban areas (Figure 3).

### 3.4. Relationship between Built-Up Environment and Hypertension

The results of hierarchical and multilevel generalized structural equation model of hypertension are shown in Figure 4, which lists the influence of four types of built-up environments on the prevalence of hypertension through six potential pathways.

#### 3.4.1. Public Administration and Service Facilities

There is a significant negative correlation between the density of public management and service facilities (β = −0.42, 95% CI −0.48–0.35) and the prevalence of hypertension (Figure 4a), which shows that the denser such facilities are, the lower the prevalence of hypertension. There is a significant positive correlation between the density of public management and service facilities and the frequency of activities (β = 0.21, 95% CI 0.14–0.30), which indicates that the richer the resources of such facilities, the higher the frequency of residents’ activities—that is, residents will move more in places where public management and service facilities are dense. There is a significant negative correlation between the density of public management and service facilities and PM_2.5_ concentration (β = −0.11, 95% CI −0.19–0.03), which indicates that PM_2.5_ concentration is relatively low and air quality is good in places with dense facilities. There is no significant correlation between the density of public management and service facilities and NDVI (β =−0.07, 95% CI −0.04–0.15)—that is, there is no correlation between the density of such facilities and the degree of vegetation coverage. To sum up, the functionality and high accessibility of public management and service facilities can enhance residents’ willingness to travel and increase the frequency of outgoing activities, and the air quality around the facilities is good, having a positive impact on human health. Residents who are close to schools, hospitals, gymnasiums and other public facilities are likely to do walking, cycling and take active travel modes to enhance their physical activity intensity, while residents living far away would reduce their travel frequency and take less physical activity. Combined with residents’ physical activity intensity, air quality and the impact of facilities on residents, it can be inferred that the prevalence of hypertension is relatively low in communities with abundant public management and service facilities.

#### 3.4.2. Business Services Facilities

There is a significant positive correlation between the facility density of business service industry (β = 0.24, 95% CI 0.04–0.58) and the prevalence of hypertension (Figure 4b)—that is, the prevalence of hypertension is relatively high in places with dense facilities. There is a significant positive correlation between the density of commercial service facilities and the frequency of activities (β = 0.05, 95% CI −0.34–0.49), which indicates that the richer the resources of such facilities, the higher the frequency of residents’ activities—that is, residents will be attracted to go out by commercial service facilities. There is no obvious correlation between the density of commercial service facilities and PM_2.5_ concentration (β = 0.09, 95% CI 0.18–0.50) and NDVI (β = −0.13, 95% CI −0.49–0.38)—that is, there is no clear correlation between the density of such facilities and air quality or green index. To sum up, residents spontaneously gather in leisure places such as restaurants, shopping centers or cinemas in their spare time. The lower the time and space cost of reaching such facilities is, the higher the travel frequency of residents has. Therefore, in communities rich in commercial service facilities and resources, residents go out frequently and have greater physical activity intensity, but some negative effects may be caused, such as noise. The closer they are to shopping malls, supermarkets and restaurants, the more convenient they can get food, which increases residents’ extra calorie intake and offset or even exceeds the benefits brought by physical activity. Combined with the impact of physical activities and facilities on residents, it can be judged that the prevalence of hypertension is relatively high in places with dense facilities in commercial services.

#### 3.4.3. Roads and Transport Facilities

There is a significant positive correlation between the density of roads and transportation facilities (β = 0.46, 95% CI 0.37–0.53) and the prevalence of hypertension (Figure 4c)—that is, the prevalence of hypertension is relatively high in places with dense road transportation facilities. There is a positive correlation between the density of roads and traffic facilities and PM_2.5_ concentration (β = 0.21, 95% CI 0.11–0.31) and activity frequency (β = 0.12, 95% CI 0.01–0.22)—that is, the air pollution and residents’ going out frequency are relatively high in places with dense facilities. However, there is no data showing that there is a significant relationship between the density of roads and traffic facilities and NDVI (β = −0.02, 95% CI −0.15–0.11)—that is, there is no correlation between such facilities and greening degree. To sum up, the data show that bus stations, subway stations and places crisscrossing roads aggravate air pollution due to excessive automobile exhaust emissions; As a necessary point for residents to travel, communities with large flow of people, high population density and convenient transportation where roads and transportation facilities are located can promote residents’ travel and enhance their activity intensity. Combined with the impact of air quality, physical activity and facilities on residents, it can be judged that places with dense roads and transportation facilities will increase the prevalence of hypertension.

#### 3.4.4. Industrial Facilities

There is a significant positive correlation between the density of industrial facilities (β = 0.53, 95% CI 0.46–0.60) and the prevalence of hypertension (Figure 4d)—that is, the prevalence of hypertension is relatively high in places with dense industrial facilities. There is a significant positive correlation between the density of industrial facilities and PM_2.5_ concentration (β = 0.22, 95% CI 0.13–0.31)—that is, where such facilities are dense, PM_2.5_ concentration is higher and air quality is worse. There is a significant negative correlation between the density of industrial facilities and the frequency of activities (β = −0.13, 95% CI −0.25–0.00)—that is, the frequency of residents going out will decrease in places with dense facilities. In addition, there is no evidence that there is a correlation between the density of industrial facilities and NDVI (β = 0.03, 95% CI −0.13–0.10)—that is, there is no obvious linear correlation between such facilities and greening degree. To sum up, the data show that, due to the emission of air pollutants such as industrial waste gas, communities rich in industrial facilities and resources have relatively more severe air pollution. Additionally, factories, workshops and other facilities are generally built far away from the city center, the surrounding public service facilities, commercial facilities and road traffic facilities are relatively few, and life is inconvenient. Therefore, residents may reduce the frequency of going out or take travel tools with less physical activity to their destinations. Combined with the impact of air quality, physical activity and facilities on residents, it can be judged that the prevalence of hypertension is relatively high in communities rich in industrial facilities.

Based on the above model results, it is found that the relationship between NDVI, PM_2.5_ concentration and activity frequency is relatively fixed, and there is a significant negative correlation between NDVI and PM_2.5_ concentration, but the existing results do not show the correlation between NDVI and activity frequency, while PM_2.5_ concentration is significantly negative correlation with activity frequency. Community greening indexes affect air quality and residents’ activity frequency. When the greening degree around the community is higher and the air quality is relatively good, residents will integrate the environment and air quality, increasing the frequency of going out and enhancing the intensity of physical activity. As for the direct influence of PM_2.5_ concentration and activity frequency on the prevalence of hypertension, the results show that both PM_2.5_ concentration and activity frequency have an impact on the prevalence of hypertension. According to all models, there is a significant positive correlation between PM_2.5_ concentration and hypertension prevalence, while there is a significant negative correlation between activity frequency and hypertension prevalence. Therefore, it can be inferred that high air pollution increases the probability of hypertension; higher activity frequency and greater activity intensity make the prevalence of hypertension relatively smaller.

This paper summarizes the direct, indirect and overall effects of different types of built-up environments on the prevalence of hypertension in the sequence mediation model (Table 3). It is found that commercial service facilities, road and transportation facilities and industrial facilities have significant positive direct effects on the prevalence of hypertension, while public management and service facilities have significant negative direct effects on the prevalence of hypertension. This may indicate that business services, roads and transportation and industrial facilities may have a direct effect on increasing the prevalence of hypertension, while public management and service facilities may have a direct effect on reducing the prevalence of hypertension. At the same time, public management and service facilities (β = −0.08, 95% CI −0.12–0.05) and commercial service facilities (β = −0.02, 95% CI −0.24–0.20) also have significant negative indirect effects on the prevalence of hypertension, while road and transportation facilities (β = −0.13, 95% CI −0.23–0.02) and industrial facilities have significant positive indirect effects on the prevalence of hypertension. When exploring the action path of built environment on the prevalence of hypertension, it is found that public management and service facilities may have a negative impact on the prevalence of hypertension through PM_2.5_ concentration, activity frequency and PM_2.5_ concentration–activity frequency path. Commercial service facilities may have a negative impact on the prevalence of hypertension through the activity frequency path. Roads, transportation facilities and industrial facilities may have a positive impact on the prevalence of hypertension through PM_2.5_ concentration, activity frequency and PM_2.5_ concentration–activity frequency path. This may indicate that public management and service facilities may reduce the prevalence of hypertension by reducing PM_2.5_ concentration, increasing the frequency of residents’ activities and the joint action of two intermediary variables. Commercial service facilities can only reduce the prevalence of hypertension by increasing the frequency of residents’ activities. Roads, transportation facilities and industrial facilities can increase the prevalence of hypertension by increasing PM_2.5_ concentration, reducing the frequency of residents’ activities and the combined action of two intermediary variables.

According to the research results, public management and service facilities, commercial service facilities, roads and transportation facilities and industrial facilities all have the most significant influence on the prevalence of hypertension through activity frequency paths (75.00%, 100.00%, 66.67%, 50.00% and 60.00%).

### 3.5. Sensitivity Analysis of Statistical Results of Different Life Circles

According to the above research results, there is a significant relationship between public management and service facilities, commercial service facilities, road and transportation facilities and industrial facilities and the prevalence of hypertension at the community level. In order to explore at different research scales whether there is any difference between the size and direction of the influence of built environment on the prevalence of hypertension and the original research scale, therefore, according to the activity range corresponding to the five-minute, ten-minute and fifteen-minute living circles divided in the Planning and Design Standard of Urban Residential Areas GB50180-2018, buffer areas with radii of 300 m, 500 m and 800 m are established, in turn, by using the buffer analysis tool in ArcGIS10.2, and the sensitivity analysis results are calculated according to the Meta analysis tool in STATA16 (Figure 5). When exploring different research scales, public management and service facilities (Figure 5a), commercial service facilities (Figure 5b) and roads and transportation facilities (Figure 5c) no longer have significant effects on the prevalence of hypertension in the 800 m buffer zone. The possible reason is that the influence scope of these three types of facilities is limited, and residents would choose the target facilities that are close to each other independently. When the distance reaches 800 m, the target facilities no longer have obvious influence on the prevalence rate of hypertension. Therefore, the effect of these three types of facilities on the prevalence rate of hypertension is affected by the range of residents’ activities; Industrial facilities (Figure 5d) are slightly different from the other three types of facilities. At the community level, industrial facilities have a significant impact on the prevalence of hypertension. There is no significant effect in 300 m and 800 m buffer zones, while the influence in 500 m buffer zone is significant, which shows that the influence of industrial facilities on the prevalence rate of hypertension is irregular in different buffer ranges, due to the fact that influence range of industrial facilities on the prevalence rate of hypertension is far larger than the real range of residents’ daily activities; and the statistical results based on different buffer ranges have no influence on the significance of this model. In addition, the correlation direction and significance between hypertension prevalence and built environment in other statistical results are basically consistent with the statistical results based on community level; that is to say, considering the change of residents’ daily activities, the statistical results are roughly the same as those of the original research scale, which shows that the statistical results at the community level have high stability.

### 3.6. Hierarchical Analysis of Individual Characteristics Differences

Through stratified analysis, the paper explores whether the differences of individual age and sex between different types of built-up environment and the prevalence of hypertension can affect the results. After adjusting for covariates, the built-up environment still has a significant impact on the prevalence of hypertension. The results show that the relationship between built-up environment and hypertension prevalence is obviously different due to different ages and genders (Figure 6). In the road and traffic facility model (Figure 6c), the correlation between male and female results is similar. Therefore, there is little difference between men and women in the impact of roads and transportation facilities on the prevalence of hypertension. In the models of public administration and service facilities (Figure 6a), business service facilities (Figure 6b) and industrial facilities (Figure 6d), the results of men are more correlated than those of women, so men are more vulnerable to these three types of built environments. At the age level, the elderly may go to scenic spots, sports parks, hospitals, schools and other related public places with more frequency for a series of activities such as walking, seeing a doctor, and picking up and dropping off children to and from school. Therefore, residents aged 60 and over are more vulnerable to public management and service facilities; In the models of business service facilities, roads and transportation facilities and industrial facilities, residents under 60 years old have stronger correlation. This may indicate that, according to the activity preference, daily travel intensity and work nature of residents under 60 years old, most of the residents, who choose to patronize shopping centers, theaters, KTV and other leisure and entertainment places, travel by convenient means of transportation such as buses and subways, and choose to work in factories are non-elderly people. Therefore, residents under 60 years old are more vulnerable to these three types of built environments.

## 4. Discussion

### 4.1. Main Research Results

As far as we know, this is the first-time study carried out in a heavy industry city. The study collected health samples of residents in four main urban areas of Lanzhou and found that there is a correlation between the built environment of different functional types and the prevalence rate of hypertension. The findings are as follows: public management and service facilities indirectly lower the prevalence rate of hypertension by reducing PM_2.5_ concentration and increasing activity frequency; commercial service facilities indirectly reduce the prevalence of hypertension with increasing the frequency of residents’ activities; roads and transportation facilities indirectly lead to an increase in the prevalence of hypertension owing to the increasing concentration of PM_2.5_ and the increasing frequency of activities and industrial facilities can increase the concentration of PM_2.5_ and reduce the frequency of activities, which indirectly leads to an increase in the prevalence of hypertension. In terms of the overall effect, the establishment of public management and service facilities can effectively mitigate the risk of hypertension among community residents, a healthy and green environment can effectively reduce the concentration of air pollutants and the formation of a vibrant environment will increase the frequency of community residents’ outings, making it easier for life to enter a good cycle of health and positivity, which will have excellent benefits in reducing the probability of hypertension. On multiple research scales, sensitivity tests of 300 m, 500 m and 800 m buffer zones in four functional built-up environments were carried out. It is found that the influence of public management and service facilities, commercial service facilities and road and transportation facilities on the prevalence of hypertension is related to the range of residents’ activities, while the influence of industrial facilities on the prevalence of hypertension has no relation to the range of residents’ activities, and there is not much difference in significance, which shows that the feasibility and stability of the original calculation based on the community level are high, and the statistical results present high credibility. In addition, after stratifying the gender and age of the sample, it is found that the correlation between business services and industrial facilities and the prevalence of hypertension is stronger among men and residents under 60 years old.

### 4.2. Existing Evidence of Environmental Impact on Hypertension

Consistent with previous studies, chronic noncommunicable diseases were not only affected by genetic factors and individual behaviors, but the accumulation of negative factors in the built environment also led to a growth in the incidence of diseases [7,47]. As Zhang Yanji’s research results showed, places with dense roads and transportation facilities were not conducive to cultural and sports activities and green travel, which led to an increase in the prevalence of chronic diseases such as hypertension, but more public management and service facilities such as park attractions effectively reduced the incidence of hypertension [31]. A study from Canada implied that more availability of parks fired residents’ activity enthusiasm, thus reducing the risk of hypertension 4. However, there are relatively few studies on the influence of green space on hypertension, and the results are inconsistent. A study from Shanghai found that although green space could promote the social interaction between middle-aged and elderly people, there was no correlation between its accessibility and coverage and the prevalence of hypertension [48]. A Belgian study showed that there was a negative correlation between community green exposure and adult blood pressure [22]. However, this paper did not get the result that NDVI could affect the prevalence of hypertension through PM_2.5_ concentration and activity frequency.

Most of the early studies started from the 5D of the built environment to explore the impact on chronic noncommunicable diseases [11,15,17,49,50]. Different from these previous studies, this paper explores the comprehensive impact of different facilities on the prevalence of hypertension from the functional types on the basis of considering the density and accessibility of the built environment. It is concluded that the prevalence rate of hypertension is relatively low in communities with abundant public management and service facilities. However, the prevalence of hypertension is relatively higher in communities rich in commercial service facilities, roads and transportation facilities and industrial facilities. It may be ascribed to the relatively good air quality and high frequency of residents’ activities in places rich in public facilities such as parks and gymnasiums, which can effectively inhibit the risk of hypertension. Although commercial service facilities, roads and transportation facilities can stimulate the frequency of residents’ outings, their negative effects also increase the occurrence of hypertension. Communities with abundant industrial facilities increase the prevalence of hypertension; that is, more air pollution and lower frequency of residents’ activities cause more serious the negative impact on blood pressure. Some certain results are different from early ones. For example, Yu Yifan et al.‘s research showed that dense roads and transportation facilities had positive effects on the physical and mental health of the elderly, so the prevalence of hypertension was low [48]. The health status of community residents is not only determined by genetics or their genes but also influenced by the environment they live in at all times. When the living space is occupied by industrial facilities, traffic and road facilities and other built environment facilities that have a negative effect on the natural environment, the air quality and safety problems become more serious, and the space for recreation shrinks, so the functional facilities also affect human health negatively, increasing the probability of chronic diseases such as cardiovascular diseases, diabetes and even mental diseases. Therefore, proper planning of built environment facilities is the key to building a green and healthy city.

### 4.3. Potential Mechanism

The built-up environment around a community may influence the prevalence of hypertension by exerting impacts on physical activity, frequency of travel, air quality and vegetation coverage [2,9,31]. For example, previous studies showed that exposure to air pollutants such as PM_2.5_ and PM_10_ led to an increase in the prevalence of hypertension [3,5,17]. In rural areas, there was a significant negative correlation between community greening rate and prevalence rate, but no correlation between them in urban areas [36]; Moderate physical activity had a lower incidence of hypertension than low physical activity and high physical activity—that is, the proper frequency of outdoor activities helped reduce the risk of chronic diseases [51]. Setting three mediation variables (NDVI, PM_2.5_ concentration and Activity frequency) and six potential paths (built environment → PM_2.5_ concentration → hypertension prevalence rate, Built environment → PM_2.5_ concentration → activity frequency → hypertension prevalence rate, Built environment → NDVI → PM_2.5_ concentration → hypertension prevalence rate, built environment → NDVI → activity frequency → hypertension prevalence rate, built environment → NDVI → PM_2.5_ concentration → activity frequency → hypertension prevalence rate, built environment → activity frequency → hypertension prevalence rate), this study relates the built environment with hypertension prevalence rate. According to the results of MGSEM, the influence paths of different built environments on the prevalence of hypertension are not the same. Public management and service facilities, roads and transportation facilities and industrial facilities can all affect the prevalence of hypertension through PM_2.5_ concentration, activity frequency and PM_2.5_ concentration–activity frequency path. However, commercial service facilities can only affect the prevalence of hypertension based on the frequency of activities. This may be because the functional types and density of different facilities have different impacts on the attractiveness of community residents and air quality, so different types of built environments have different impacts on the prevalence of hypertension.

These findings are partially same as those of a previous study conducted in Nanjing, China. That is, the built environment indeed affected residents’ activities and air quality. Areas with concentrated functions, convenient transportation and high coverage of green water reduced the prevalence of hypertension because of the improvement in residents’ activities and less air pollution, and the losses and benefits caused by air pollution and physical activity to residents offset each other, which had the final effect on the prevalence of hypertension [18]. All these studies confirmed that the frequency of residential activity and air quality indeed mediated the relationship between built environment and the prevalence of hypertension. However, the results of this study did not prove that the NDVI–activity frequency pathway and NDVI-PM_2.5_ concentration–activity frequency pathway could mediate the relationship between built environment and hypertension prevalence, which is inconsistent with previous research results [24,36,52].

### 4.4. Advantages and Limitations of Research

The advantages of this study are as follows: (1) By taking the statistical blood pressure consultation data of streets and community health service stations in four main urban areas of Lanzhou in 2020 as samples, the data is new and the sample size is huge, which can improve the accuracy and accuracy of the results. (2) Compared with previous studies, this paper puts multilevel generalized structural equation model into use to analyze the influence of a built-up environment on the prevalence of hypertension and evaluates the role of PM_2.5_ concentration, activity frequency and NDVI through sequence mediation model in order to increase the robustness of the results. (3) Based on the POI data of Lanzhou City, the built environment is subdivided in terms of functions. Taking the community as the research scale to explore the corresponding contributions of different paths makes up for the simplification of influencing factors of hypertension prevalence and the shortage of single built-up environment structure in the existing research and provides a reference for optimizing urban spatial structure and rational use of urban space.

There are also some limitations in this study: (1) Only a few natural attributes of population (gender and age) are included in the research, short of social attributes of population and community characteristics in the collected hypertension data, so it is impossible to control the covariates at individual level and regional level to analyze the samples hierarchically, and thus, it is unlikely to calculate the mixed influence of covariates on the action path. (2) Residents’ preference on communities may affect the relationship between built environment and hypertension prevalence. For example, residents with chronic diseases may prefer communities with rich medical resources; as a result, the impact of educational facilities on hypertension prevalence may be overestimated. (3) On studying the influence of built environment on residents’ activity frequency, only the relationship between destination distance and travel mode is taken into consideration, while other factors affecting physical activity are not seriously discussed.

## 5. Conclusions

This study shows that different functional types of built environment do have a certain impact on the prevalence of hypertension. The paper finds that communities with dense public management and service facilities may inhibit the risk of hypertension, while in communities with abundant commercial service facilities, roads and transportation facilities and industrial facilities, the risk may increase. The data also show that PM_2.5_ concentration may inhibit the relationship between built-up environment and hypertension prevalence, but the activity frequency may promote it. However, it cannot be proven that the NDVI can regulate the relation between a built-up environment and hypertension prevalence. Through the analysis of gender and age stratification of samples, it is found that the built environment has a significant impact on the prevalence of hypertension from its own functionality and density, and the community, as the smallest unit in residents’ daily life can maximize the impact of the internal and surrounding environment on residents. This paper holds that the concept of “people-oriented” should be upheld in communities’ planning and construction, optimizing the functional composition of the built environment, rationally planning the distance between buildings and reducing the adverse effects on residents’ lives. Future research should combine the macroscopic built environment with the spatial and temporal behaviors of residents, and studies should analyze the mechanisms of the built environment on hypertension at different spatial and temporal scales from a more microscopic and comprehensive perspective. We should understand the differential characteristics of natural, human and social factors in different regions from detailed aspects to propose more regionally targeted prevention and treatment programs and public health policies for chronic diseases such as hypertension.

## Figures and Tables

**Figure 1 ijerph-20-00743-f001:**
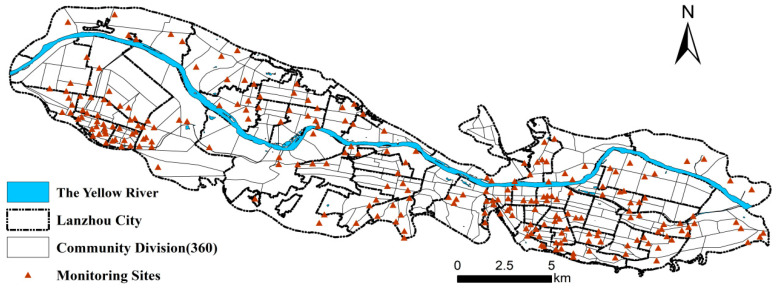
Overview of the study area.

**Figure 2 ijerph-20-00743-f002:**
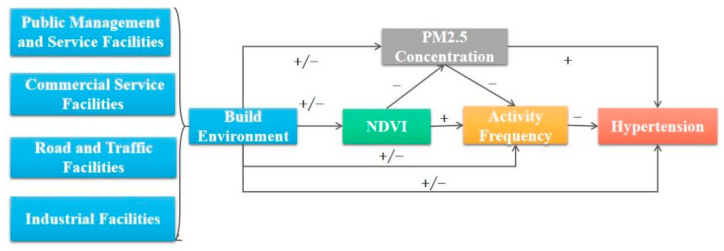
Conceptual diagram of serial mediation models linking built environment with hypertension. Note: “+” indicates positive connection and “−” indicates negative connection.

**Figure 3 ijerph-20-00743-f003:**
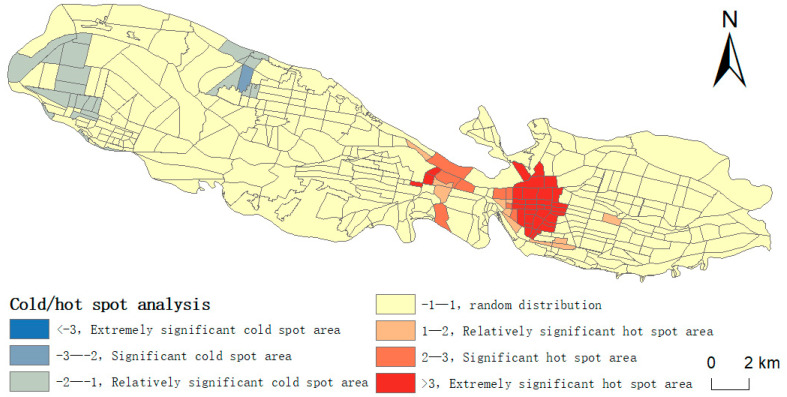
The distribution of cold and hot spots in the prevalence of hypertension in the main urban areas of Lanzhou in 2020.

**Figure 4 ijerph-20-00743-f004:**
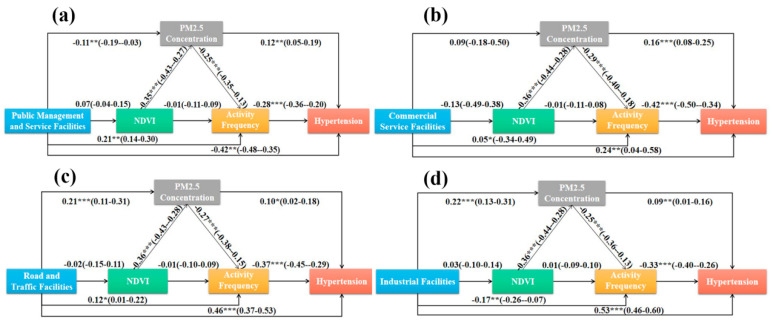
Conceptual diagram of serial mediation models linking built environment with hypertension. Note: ① (**a**): pathways of public management and service facilities on hypertension; (**b**): pathways of commercial service facilities on hypertension; (**c**): pathways of road and traffic facilities on hypertension; (**d**): pathways of industrial facilities on hypertension. ② The linear regression coefficient (β) of hierarchical multilevel structural equation in 95% confidence interval, the significance level: * *p* < 0.05, ** *p* < 0.01 and *** *p* < 0.001.

**Figure 5 ijerph-20-00743-f005:**
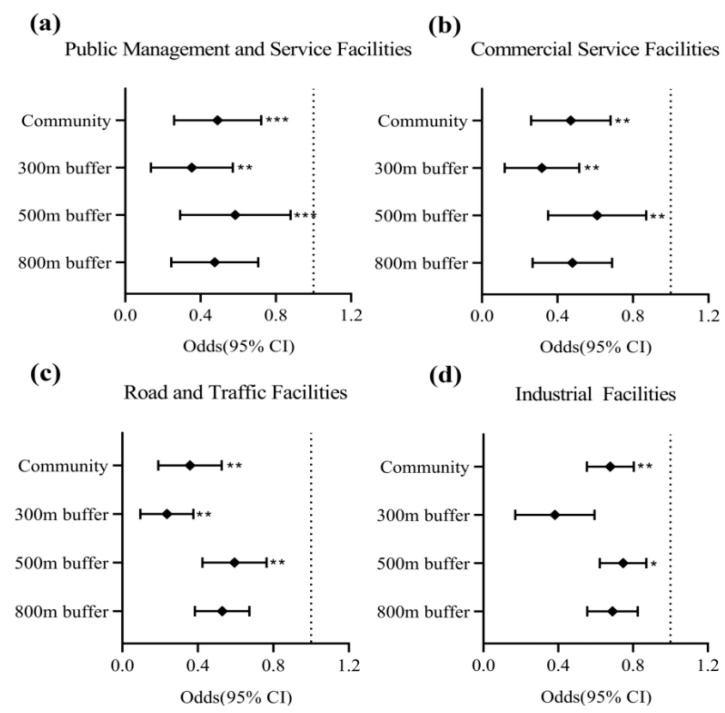
Results of sensitivity tests with the MGSEMs on the association between different types of built environments and hypertension. Note: ① (**a**): sensitivity analysis of public management and service facilities in different living circles; (**b**): sensitivity analysis of commercial service facilities in different living circles; (**c**): sensitivity analysis of road and traffic facilities in different living circles; (**d**): sensitivity analysis of industrial facilities in different living circles. ②Symbols denote probability, and bars denote 95% CI. Significance levels: * *p* < 0.05, ** *p* < 0.01 and *** *p* < 0.001.

**Figure 6 ijerph-20-00743-f006:**
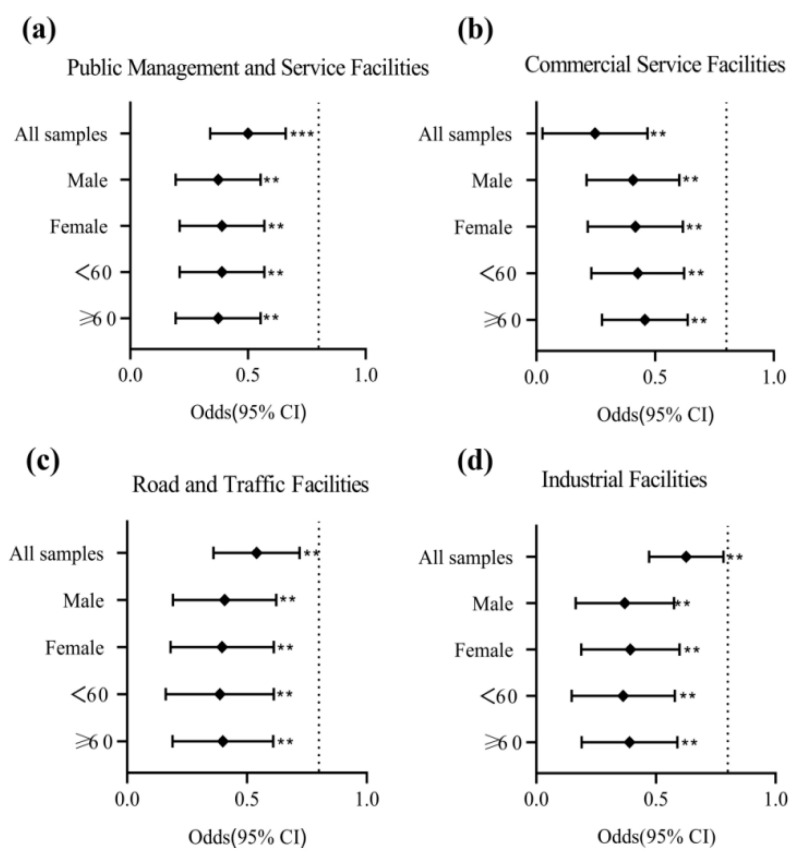
Results of MGSEM’s hierarchical analysis of the relationship between different types of built environments and hypertension. Note:① (**a**): effects of public management and service facilities on hypertension in different characteristic groups; (**b**): effects of commercial service facilities on hypertension in different characteristic groups; (**c**): effects of road and traffic facilities on hypertension in different characteristic groups; (**d**): effects of industrial facilities on hypertension in different characteristic groups; ② Symbols denote probability, and bars denote 95% CI. Significance levels: * *p* < 0.05, ** *p* < 0.01 and *** *p* < 0.001.

**Table 1 ijerph-20-00743-t001:** Deprivation index system of residential environment setting.

	Lanzhou City	Chengguan District	Anning District	Qilihe District	Xigu District
Number of patients (persons) ^a^	103,877	56,207	12,760	21,023	13,887
Age (%) ^a^					
<60	34.17	34.38	33.74	36.36	30.39
≥60	65.83	65.62	66.26	63.64	69.61
Gender (%) ^a^					
Male	41.20	41.41	41.35	40.38	41.46
Female	58.80	58.59	58.65	59.62	58.54
Systolic blood pressure (mmHg) ^a^	132.76	133.30	131.86	132.19	132.29
Diastolic pressure (mmHg) ^a^	79.90	80.11	78.55	81.77	77.54
BMI (%) ^a^					
<18.5	1.37	1.25	1.35	1.02	2.35
18.5~24.9	51.03	51.45	48.15	51.78	50.89
≥25	47.60	47.30	50.50	47.20	46.76
Population density (people/km^2^) ^a^	106	127	69	143	87
PM_2.5_ concentration (μg/m^3^) ^a^	38.48	38.56	31.80	37.84	45.72
NDVI (Median IQR) ^a^	0.61 (0.60)	0.60 (0.59)	0.61 (0.62)	0.61 (0.62)	0.61 (0.59)
Built environmental density (100 million/km)					
Public management and service facilities ^b^	756,048.42	1,350,121.11	456,349.92	620,872.00	596,850.66
Business Services Facilities ^b^	6,474,189.07	12,434,783.57	3,223,387.62	5,454,234.17	4,784,350.91
Roads and transport facilities ^b^	256,588.04	439,345.30	153,333.04	220,177.25	213,496.57
Industrial Facilities ^b^	9243.65	11,116.81	7129.96	9395.33	9332.48
Thermodynamic value (person/ha) ^a^	44.54	53.80	37.80	45.40	41.16

Note: ① ^a^ represents the statistical results of 103,877 samples and ^b^ represents the statistical results of 360 community levels. ② According to the World Health Organization standard, BMI < 18.5 is lean, BMI between 18.5 and 24.9 is normal and BMI ≥ 25 is overweight.

**Table 2 ijerph-20-00743-t002:** Appropriate indicator values for the structural equation model.

	X^2^	X^2^/df	GFI	AGFI	RMSEM	NNFI	IFI	CFI
Recommended value	The smaller the better	<3.00	>0.90	>0.80	<0.08	>0.90	>0.90	>0.95
Public management and service facilities	367.26	6.61	0.99	0.89	0.02	0.98	0.98	0.98
Business service facilities	197.97	4.19	0.99	0.93	0.07	0.97	0.98	0.98
Roads and transport facilities	295.28	2.56	0.99	0.95	0.06	0.99	0.99	0.99
Industrial facilities	354.99	1.01	0.99	0.98	0.01	0.99	1.00	1.00

Note: X^2^, X^2^/df, GFI, AGFI, RMSEM, NNFI, IFI and CFI are chi-square value, chi-square degree of freedom ratio, goodness-of-fit index, adjusted goodness-of-fit index, mean square and square root of asymptotic residuals, standardized fitting index, multiplication fitting index and comparative fitting index, respectively.

**Table 3 ijerph-20-00743-t003:** Association between built environment types and the prevalence of hypertension: serial mediation model.

Path		M1	M2	M3	M4	M5	M6	M7
Public management and service facilities	Direct effect							−0.42 ** (−0.48–−0.35)
	Indirect effect	−0.01 ** (−0.03–0.00)	−0.01 ** (−0.02–0.00)	0.00 (−0.01–0.00)	0.00 (−0.01–0.00)	0.00 (0.00–0.00)	−0.06 *** (−0.09–−0.04)	−0.08 *** (−0.12–−0.05)
	Overall effect							−0.50 *** (−0.56–−0.44)
Business service facilities	Direct effect							0.24 ** (0.04–0.58)
	Indirect effect	0.02 (−0.03–0.09)	0.01 (−0.02–0.07)	0.01 (−0.02–−0.04)	0.01 (−0.02–0.03)	0.00 (−0.02–0.01)	−0.02 * (−0.20–0.14)	−0.02 * (−0.24–0.20)
	Overall effect							0.22 ** (0.004–0.58)
Roads and transport facilities	Direct effect							0.46 *** (0.37–0.53)
	Indirect effect	0.02 ** (0.00–0.05)	0.02 *** (0.01–0.04)	0.00 (0.00–0.01)	0.00 (0.00–0.01)	0.00 (0.00–0.00)	0.04 * (0.01–0.08)	0.08 ** (0.04–0.13)
	Overall effect							0.54 ** (0.46–0.62)
Industrial facilities	Direct effect							0.53 *** (0.46–0.60)
	Indirect effect	0.02 * (0.00–0.04)	0.02 *** (0.01–0.03)	0.00 (−0.01–0.01)	0.00 (−0.01–0.01)	0.00 (0.00–0.00)	0.06 *** (0.03–0.09)	0.10 *** (0.06–0.14)
	Overall effect							0.63 ** (0.59–0.68)

Note: The linear regression coefficient (β) of hierarchical multilevel structural equation in 95% confidence interval, the significance level: * *p* < 0.05, ** *p* < 0.01 and *** *p* < 0.001. M1: built environment → PM_2.5_ concentration → hypertension prevalence rate. M2: built environment → PM_2.5_ concentration → activity frequency → hypertension prevalence rate. M3: Built Environment → NDVI → Activity Frequency → Prevalence of Hypertension. M4: Built environment → NDVI → PM_2.5_ concentration → hypertension prevalence rate. M5: Built environment → NDVI → PM_2.5_ concentration → Activity frequency → Prevalence of hypertension. M6: Built Environment → Activity Frequency → Prevalence of Hypertension. M7: Built environment → Prevalence of hypertension.

## Data Availability

Not applicable.

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
