# Peer review of "Relationship between Built-Up Environment, Air Pollution, Activity Frequency and Prevalence of Hypertension—An Empirical Analysis from the Main City of Lanzhou"

_ijerph, 2022, doi:10.3390/ijerph20010743_

Round 1

Reviewer 1 Report

This study presents a case study of the relationship between Built-up Environment, Air Pollution, Activity Frequency and Prevalence of Hypertension,an Empirical Analysis from the Main City of Lanzhou, China, the topic is interesting and practical, I recommend the authors to address the following issues:

1. the abstract should be well structured. The structure of the abstract of this article is incomplete.

2. The conclusions stated in this article are not distinguished from the findings. Conclusions are different from results and findings. Results and findings are specific in nature, while conclusions are more general in nature. The conclusion is the result of upgrading and deducing from the results and findings. In the abstract, results are results and conclusions are conclusions. In the conclusion section of a paper, the

3. the discussion in this article is not fully structured. If others have done similar studies, you need to explain which aspects of your study are consistent with others, those aspects are different, and why they are different. If the author has prior research, you should also explain how this article is an improvement.

Finally, there are repetition and colloquialism problems in individual text expressions, and the central sentence of the paragraph is not obvious, which is suggested to be further improved.

Reviewer 2 Report

Article title:

“Relationship between Built-up Environment, Air Pollution, Activity Frequency and Prevalence of Hypertension—An Empirical Analysis from the Main City of Lanzhou”

Keywords:

built environment; hypertension; pathways of action; multilevel generalized structural equation model; healthy public policy

Comments:

Undoubtedly, the topic is of particular interest because it links the health of residents with the built environment.

The research is well-documented, contains correct material, and makes effective use of the numerous previous studies in this topic. In terms of the research's structure. The abstract is well-written and accurately represents the purpose, methodology, and findings. The opening effectively introduces the topic and identifies the research issue. The methodology is also quite straightforward.

Although the research is well supported by data and statistics, however discussing the built environment requires additional support in the form of land use plans and images. These can be used to demonstrate the differences in the built environment and explain in a more concrete manner the urban density and the lack of public service spaces.

Reviewer 3 Report

The paper deals with an interesting topic, namely the link between the built environment in different typologies and medicine, in line with recent international research aimed at the well-being of citizens. 

The text is well-written and structured, the methodology is clear, and the data is complete. 

Some suggestions:

- better define what future developments might be.

- clarify whether this method can also be applied to other contexts. 

In general, the paper is worthy of publication.
